# The Role of the Estrogen-Related Receptor Alpha (ERRa) in Hypoxia and Its Implications for Cancer Metabolism

**DOI:** 10.3390/ijms24097983

**Published:** 2023-04-28

**Authors:** Leslie Chaltel-Lima, Fabiola Domínguez, Lenin Domínguez-Ramírez, Paulina Cortes-Hernandez

**Affiliations:** 1Segal Cancer Center, Lady Davis Institute for Medical Research, Jewish General Hospital, Montreal, QC H3T 1E2, Canada; leslie.chaltellima@mail.mcgill.ca; 2Division of Experimental Medicine, McGill University, Montreal, QC H4A 3J1, Canada; 3Centro de Investigación Biomédica de Oriente (CIBIOR), Instituto Mexicano del Seguro Social (IMSS), Atlixco 74360, Mexico

**Keywords:** ERR, HIF-independent response to hypoxia, cancer, metabolic adaptation to hypoxia, VEGF, angiogenesis, ischemia, PGC-1α

## Abstract

Under low oxygen conditions (hypoxia), cells activate survival mechanisms including metabolic changes and angiogenesis, which are regulated by HIF-1. The estrogen-related receptor alpha (ERRα) is a transcription factor with important roles in the regulation of cellular metabolism that is overexpressed in hypoxia, suggesting that it plays a role in cell survival in this condition. This review enumerates and analyses the recent evidence that points to the role of ERRα as a regulator of hypoxic genes, both in cooperation with HIF-1 and through HIF-1- independent mechanisms, in invertebrate and vertebrate models and in physiological and pathological scenarios. ERRα’s functions during hypoxia include two mechanisms: (1) direct ERRα/HIF-1 interaction, which enhances HIF-1′s transcriptional activity; and (2) transcriptional activation by ERRα of genes that are classical HIF-1 targets, such as VEGF or glycolytic enzymes. ERRα is thus gaining recognition for its prominent role in the hypoxia response, both in the presence and absence of HIF-1. In some models, ERRα prepares cells for hypoxia, with important clinical/therapeutic implications.

## 1. Introduction

Oxygen, the final electron acceptor of the mitochondrial respiratory chain for ATP production, is crucial for all aerobic organisms. Despite the complex systems that higher organisms have developed to irrigate every organ and constantly provide all cells with oxygen, a variety of conditions can limit oxygen levels: some pathological (ischemia, tumor development, anemia, lung disease) and some physiological (embryonic development, exercise). Hypoxia is thus defined as a decrease in the oxygen supply to levels insufficient for cellular function [1]. The precise O_2_ concentration that represents hypoxia varies from tissue to tissue, and likely even between individuals, as different tissues are exposed to different physiological oxygen concentrations (termed “physioxia”), most ranging from 3 to 9% oxygen (23–70 mmHg) [1]. O_2_ determinations suggest that most mammalian high-energy-demand tissues, such as the brain, muscle, liver, renal cortex and heart, maintain physiological oxygen concentrations between 2.5 and 5.5% [1,2]. Since hypoxia can quickly become life threatening, it triggers a response to modulate blood flow, change energy metabolism, induce angiogenesis and cell differentiation and, ultimately, induce apoptosis [3].

The hypoxia response has a fast component that relies on existing proteins, such as ion channels and already-expressed signaling pathways, with effects such as blood redistribution, tachypnea/tachycardia, widespread inhibition of protein translation and impaired cell proliferation [4,5]. It also has a well-characterized slower component that induces the expression of about one-thousand specific hypoxia genes once hypoxia is installed [6]. Hypoxia-inducible factors (HIFs) are central transcription factors responsible for protein expression during the slow component of the hypoxia response [7], but other transcription factors also participate. One such example is the estrogen-related receptor alpha (ERRα), a ubiquitously expressed orphan nuclear receptor, abundant in high-energy-demand tissues such as the heart, kidneys and cancer cells [8,9,10,11]. This review discusses the role that ERRα plays in the hypoxia response in synergism with HIF-1 and by HIF-independent mechanisms.

## 2. HIF-1, -2 and -3 Mediate the Hypoxia Response

The HIF-mediated transcriptional response to hypoxia was discovered in the 1990s [7] and received the Nobel Prize in Physiology or Medicine in 2019 [12]. Due to its central importance to physiology and to pathological states such as cancer, HIF has been extensively studied and reviewed [6,12,13,14,15]. HIF-1 was the first such factor described and remains the most characterized, but HIF-2 and HIF-3 have been described as well [14,16]. All three, are heterodimeric basic helix–loop–helix transcription factors consisting of subunits α and β. Each α subunit is O_2_-regulated, as it is constantly targeted for destruction during normoxia via the Von Hippel Lindau protein and the E2-ligase/ubiquitin proteasome pathway [12,13,14,15,17]. In turn, the HIF-1β subunit (initially known as the aryl hydrocarbon nuclear translocator *ARNT*, UniProt P27540) is constitutively expressed [12,13,14,15] and can heterodimerize with the different oxygen-sensitive α subunits (HIF-1α, HIF-2α or HIF-3α) to create tissue-specific HIF-1, HIF-2 or HIF-3 transcription factors. While HIF-1α is conserved from Parazoa to vertebrates and expressed in most cells [13,14], HIF-2α and -3α are only present in vertebrates and expressed tissue-specifically [12,14,16].

HIF-1α (*HIF1A*, UniProt Q16665), the most characterized homolog, is an 826-residue protein that locates to the cytoplasm during normoxia where it is constantly destroyed [12,13,14,15,17]. Under hypoxic conditions or in the presence of iron chelating agents, it translocates to the nucleus and dimerizes with HIF-1β to form the functional HIF-1 complex that activates gene transcription [12,13,14,15,17]. HIF-1α DNA binding activity and stabilization have half maximal responses between 1.5 and 2% O_2_ and maximal response at 0.5% O_2_, determined in human cultured cells [2]. HIF-1 requires co-activators/transactivators such as CREB binding protein (CBP), p300 [18,19] and others [14], and the complex binds to hypoxia response elements (HREs) with the 5′-RCGTG-3′ consensus [14,20].

Around one-thousand genes have been identified as HIF-1 targets [21] and they can be grouped into two functional categories: those that increase oxygen supply to tissues and those that decrease oxygen consumption by tissues [6,15]. In the first category, HIF elicits an increase in oxygen delivery to tissues by triggering erythropoiesis and angiogenesis through the expression of erythropoietin, the hormone that controls red cell production and blood O2-carrying capacity [7], and VEGF (vascular endothelial growth factor), the main protein that stimulates new blood vessel formation [22]. In the second category (decreasing oxygen consumption) are many genes that modify energy metabolism [23,24]. For example, in hypoxia, oxidative phosphorylation (OXPHOS) is restricted due to the lack of O_2_, and cells shift to anaerobic glycolysis through the increased expression of glycolytic enzymes by HIF-1 [23]. To compensate for the much lower ATP generation per glucose molecule through glycolysis than through OXPHOS, HIF-1 activates transcription of the *SLC2A1* and *SLC2A3* genes coding the glucose transporters GLUT1 and GLUT3 that increase glucose uptake [23]. Moreover, to inhibit the conversion of pyruvate to acetyl-CoA, HIF-1 activates gene transcription to decrease pyruvate flux to the Krebs cycle and increase lactate production. Examples of activated genes are the *PDK1* gene encoding PDH kinase, which phosphorylates and inactivates the catalytic subunit of pyruvate dehydrogenase (PDH) [24], and the *LDHA* gene encoding lactate dehydrogenase A, which directly catalyzes the conversion of pyruvate to lactate [20]. In this way, the HIF-1 response to hypoxia is, in part, executed through metabolic adaptation.

To form HIF-2, HIF1-β heterodimerizes with HIF-2α (*EPAS1*, UniProt Q99814; also called endothelial PAS domain protein 1, HIF-1α-like factor (*HLF*), HIF-1α related factor (*HRF*) and member of the PAS superfamily-1 (*MOP-1*)). HIF-2α has similarities to HIF-1α in terms of domain structure, O_2_-dependent degradation, DNA sequence recognition (also binds to hypoxia response elements, HREs) and heterodimerization, yet exhibits different effects over gene expression mostly due to tissue-specific expression and kinetics [12,14,16]. The kinetics of HIF-1α and HIF-2α suggest that the former exerts a more rapid response at oxygen levels around 1–2%, whereas HIF-2α action occurs after prolonged hypoxia [25]. In contrast to HIF-1α that expresses ubiquitously, HIF-2α only expresses in certain tissues such as embryonic and adult vascular endothelia, lung, placenta, heart, renal interstitial cells and liver [16,26]. HIF-2α also has specific coactivators such as NF-κB essential modulator, and Ets1 that do not interact with HIF-1α. Genes strongly activated by HIF-2 are erythropoietin, VEGF receptor 2, insulin-like growth factor-binding protein-2 and plasminogen activator inhibitor-1 [26,27]. HIF-2 acts more effectively than HIF-1 on erythropoietin and iron metabolism genes, whereas VEGF and GLUT1 are similarly activated by HIF-1 and HIF-2, and glycolytic enzymes are more activated by HIF-1 [26,27]. Thus, within the hypoxia response that requires gene expression, HIF-1 constitutes a faster metabolic component; in turn, HIF-2 is more effective on erythropoiesis control once hypoxia persists.

In turn, HIF-3α (*HIF3A*, UniProt Q9Y2N7) can be present in different splice variants that depend upon the tissue, some of which are proposed to have negative regulatory functions on the hypoxia response [16,28]. This seems to be the case for the short HIF-3α variant that is also called inhibitory PAS domain protein (IPAS), which expresses in corneal epithelium and putatively prevents vascularization there [16,28]. Thus, HIF-3α could have specific regulatory roles that will not be further reviewed here, but that have been discussed in [16,28].

## 3. Introduction to the Estrogen-Related Receptors (ERR)

The orchestration of metabolic adaptation central for hypoxia survival seems to involve other transcription factors that specialize in the control of energy metabolism, such as the estrogen-related receptor (ERR) subfamily of nuclear receptors. Here, we describe the subfamily, with a focus on ERRα, and then analyze the evidence and mechanisms that link ERRα to the regulation of hypoxic metabolism and angiogenesis.

The ERR subfamily belongs to group III of the nuclear receptor superfamily (orphan nuclear receptors) [8,29]. In humans and most vertebrates, it comprises three members: ERRα (NR3B1), ERRβ (NR3B2) and ERRγ (NR3B3). However, only one ERR gene has been found in invertebrates such as Urochordates, *Drosophila melanogaster*, and in the mosquito *Anopheles gambiae*, but none seem to exist in *Caenorhabditis elegans* [30,31,32]. A search in the Inparanoid database (version 9) confirmed that no ERR homologs are identifiable in nematodes [33].

The general structure of ERRs is common to nuclear receptors, including four functional domains: N-terminal (NTD), DNA-binding (DBD), hinge, and a putative ligand-binding domain (LBD) [8,10,34] (Figure 1A). The DBD comprises two cysteine-rich zinc finger motifs, which are required for DNA binding and recognize the ERR response elements (ERREs), composed by the sequence TNAAGGTCA [35,36,37]. The three members of the ERR family (α, β and γ) bear high similarity, particularly in the DBD and LBD domains [38], but they have somewhat different functions and their expression is tissue-specific. ERRγ and ERRβ bear more similarity with each other than with ERRα [38]. ERRα is the most abundant member of the family, expressed in most cells, and with higher levels in those with high energy demand, especially in cells that oxidize fatty acids [35,36,37], compatible with its role in the transcriptional control of energy metabolism.

ERRα (*ESRRA*, UniProt P11474) is a 423-residue protein and the first orphan nuclear receptor identified in a 1988 screen for genes related to estrogen receptor alpha (ERα) [39], just a few years before the identification of HIF. Unlike estrogen receptors, no endogenous ligand has been described for ERRs; thus, ERRα, the first orphan nuclear receptor identified, remains among the “non-adopted” orphans [34,40]. Recently, it was reported that an endogenous 19-nor steroid estradienolone, found in the urine of pregnant women, can bind and act as an inverse agonist to ERRα and ERRγ [41]. There is still little information to discern if this could be the long-sought endogenous ligand of the family, but it seems unlikely due to the plethora of crucial functions that have been described for the ERRs and which do not require a ligand (reviewed below).

Due to ERRα’s sequence identity to ERα, particularly in the DBD and LBD (68% and 37% residue identity, respectively [39]), it was initially suggested that these two receptors shared common targets, co-regulatory proteins and sites of action [42,43]. However, through the combination of computational biology; ERRα silencing; interaction with the co-activators such as PGC-1α; DNA binding assays; chromatin immunoprecipitation with sequencing (CHIP-seq); and reporter gene approaches, the differences between ERRα’s and ERα’s functions have become apparent [8,36,37,44]. ERRα regulates a different set of genes to Erα and is not involved in estrogen response. ERRα is mainly involved in the transcriptional regulation of metabolic pathways spanning carbohydrate, lipid and amino acid metabolism, importantly through the regulation of genes for mitochondrial biogenesis, oxidative phosphorylation (OXPHOS) and fatty acid oxidation [8,10,11,37]. The other members of the ERR family also control aspects of metabolism, although in specific tissues [8,29,32,38]. Overall, ERRs occupy the promoters of over 700 genes that encode mitochondrial proteins, regulating mitochondrial biogenesis [8].

Specifically, ERRα binds to the promoters of glycolysis and tricarboxylic acid cycle (TCA) genes, and to OXPHOS genes such as ATP synthase b (*ATP5PB*), cytochrome c (*CYCS*), *COX4*, *GABPA* and adenine nucleotide translocator 1 (*ANT1*) [44]. ChIP-seq studies performed in mouse or human, liver, kidney, macrophages or cancer cell lines confirmed that ERRα can bind to promoters for OXPHOS (*SDHD* and *SUCLA2*), TCA (*FASN*), glycolysis/gluconeogenesis and lipid metabolism genes (*GPAM* and *ELOVL6*) [37,45,46,47,48]. Furthermore, ERRα activates the promoters of β-oxidation genes, such as *ACADM* (medium-chain acyl co-A dehydrogenase) and *CPT1A* (carnitine palmytoyl transferase 1A), as well as the promoters of glutamine transporters and enzymes for glutamine synthesis and catabolism [35,44]. In summary, ERRα’s function can be described as activating gene expression to adapt energy production to physiological or pathological stress. ERRα’s functions in cellular metabolism have been reviewed in [8]. In breast cancer, ERRα’s transcriptional activities mediate metabolic adaptations leading to treatment resistance [46]. The metabolic programs it controls make ERRα an ideal contributor to the hypoxia response and a potential pharmacological target.

The other members of the ERR subfamily also modulate metabolism with complementary and sometimes opposite functions to ERRα [8,29]. ERRβ has emerged as important in maintaining multipotency [38]. In breast cancer, ERRα and ERRγ seem to play opposing roles as modulators of cell metabolism: ERRγ activates TCA and OXPHOS while ERRα redirects energy metabolism to glycolysis and lactate production [8]. This balance of control is likely part of the mechanism at the core of the Warburg effect in many tumors, along with HIF-1 [8,11], but it is far from a simple on/off switch. Rather, it is a dynamic balance under tight control that is highly cell- and context-specific, where both members of the ERR family, ERRα and ERRγ, activate metabolic pathways facilitating cell survival and adaptation to the changing environment.

To bind DNA and to modulate target genes, ERRs can act as monomers, homodimers or heterodimers [36,37] (Figure 1D), although in live cells mainly homo or heterodimers have been associated to function [37,38]. ERR transcriptional activity is increased by members of the steroid receptor co-activator (SRC) family [42,49,50] and by the peroxisome proliferator-activated receptor gamma co-activator-1 (PGC-1) α and β [44,49,51] (Figure 1D). Interactions with the cofactors are mediated by ERR’s LBD, particularly by helices 11 and 12 via leucine rich motifs (H11 and H12 in Figure 1B,C), also referred to in the literature as ERR’s AF2 domain for “activation function 2” [49,52,53]. In particular, ERRα’s functions on metabolism are dependent on PGC-1α [49]. ERRα and PGC-1α influence each other’s expression, and both orchestrate the transcription of energy metabolism genes [37,53]. Recently, details on ERRα’s transcription initiation mechanism have been clarified. PGC1α was essential for p300 and mediator recruitment to activate transcription when ERRα acted on chromatin, whereas on naked DNA ERRα established direct contact with initiation factor TFIIH, and PGC1α did not further increase transcription [49,50]. While ERRα depends on PGC-1α to transcribe metabolic genes, ERRβ and γ can function independently of PGC-1α in stem cells and muscle [49], and other cofactors that interact with their AF2 domains, such as NCOA, replace PGC-1α [49,50].

Unlike ERα, the ERRs do not need ligands to interact with its co-activators and to bind DNA (they are constitutively active), probably because the putative ligand-binding pocket (LBP) is occupied by residue side chains in a conformation favored by the cofactors. In the empty ERRα crystal structures, the binding pocket is mainly occupied by the bulky phenolic ring of Phe232 (XRD structures number this residue as 328; however, numbering according to UniProt is used here), which corresponds to a less bulky Ala350 in ERα [41]. Despite this apparent lack of a ligand-binding pocket [34,52], synthetic compounds can inhibit ERRα’s constitutive activity; thus, they are considered ERRα’s inverse agonists (i.e., compounds with affinity and intrinsic activity on the protein). Among the first synthetic inverse agonists described for ERRα was XCT790 (reported in 2004), a thiadiazole acrylamide, which alters ERRα/PGC-1α signaling and is inactive against the rest of the ERRs and ERα [54]. Later, “compound 1a” (ciclohexilmetil-(1-p-tolil-1H-indol-3-ilmetil)-amine) and “compound 29” (4-(4-{[(5R)-2,4-dioxo-1,3-thiazolidin-5-yl]methyl}-2-methoxyphenoxy)-3-(trifluoromethyl)benzonitrile) were synthetized and have been crystallized in complex with ERRα’s LBD [55,56]. An analysis of these ERRα structures with inhibitors revealed a significantly larger ligand-binding pocket than in the empty protein, created by the rearrangement of amino acid residues F232, F286, F399 and F414 (328, 382, 495 and 510 in XRD 2PJL and 1XB7). F232 and F414 change conformation significantly when ERRα admits a ligand (Figure 1B vs. Figure 1C). In addition, these structures suggest that the presence of the inverse agonists disrupts the interaction between ERRα and PGC-1α, through the displacement of ERRα’s helix, to a position that interferes with co-activator recruitment [57,58].

## 4. Evidence of ERRα’s Participation in the Hypoxia Response

Next, we review the evidence for ERRα’s participation in the hypoxia response in models that span invertebrates and vertebrates, and physiological and pathological scenarios. These studies have led to the suggestion of HIF-dependent and independent mechanisms, including some that transcend ERRα’s central role as a metabolic coordinator during stress.

### 4.1. ERRα Induces VEGF Expression during Muscle Ischemia and Other Models

The work that first pointed to ERRα’s role in hypoxia came from the study of angiogenesis/ischemia where VEGF, a classical HIF-1 target central to angiogenesis, was discovered to also be inducible by ERRα in skeletal muscle [57]. Arany et al. first detected that ERRα’s co-activator, PGC-1α, was induced by hypoxia in vitro in various cell types, and in vivo in muscle [57]. Using a skeletal muscle ischemia model, these authors showed that transgenic animals overexpressing PGC-1α had increased angiogenesis with VEGF expression. PGC-1α/ERRα, but not other transcription factors co-activated by PGC-1α, were necessary to increase VEGF expression, through a mechanism that neither depended on HIF response elements (HREs) nor affected HIF-1 expression/stability [57]. Furthermore, conserved ERRα response elements (ERRES) were identified in the first intron of the *VEGF* gene and were recognized by PGC-1α/ERRα [57]. ERRα’s ability to induce VEGF expression and angiogenesis, as well as platelet-derived growth factor (PDGF) and Angiopoietin 2, has been confirmed in other models [58,59,60,61,62,63]. Some studies suggest that the effect does not require HIF-1 [57,58], while others suggest that HIF-1 can increase ERRα expression [60], and that, in turn, ERRα suppression can decrease HIF-1α [59].

In skeletal muscle, the alternatively spliced truncated isoforms of PGC-1α, NT-PGC-1α and PGC-1α4, induced VEGF expression by ERRα without increasing mitochondrial biogenesis [63] (Figure 2B). These PGC1α isoforms bind ERRα but not other transcription factors, such as NRF-1 and NRF-2 [52,63]. This suggests a mechanism by which the PGC-1α/ERRα axis can operate in hypoxia without increasing mitochondria (Figure 2A), which would likely be impaired in respiration due to the limited O_2_ to act as a terminal acceptor for OXPHOS. Additionally, other authors have suggested that PGC-1α could amplify intracellular hypoxia by activating mitochondrial biogenesis/OXPHOS as a mechanism to consume all remaining intracellular oxygen [64], thus precipitating hypoxia responses and stabilizing HIF-1α.

Other recent studies suggest that some ERRα’s responses to hypoxia in the skeletal muscle are dependent on HIF-1. ERRα is expressed during hindlimb muscle ischemia. Transgenic mice overexpressing ERRα in the skeletal muscle have faster revascularization with more muscle capillaries and higher artery/arteriole density after ischemia [65,66]. ERRα overexpression was also induced in C2C12 myotubes by oxygen deprivation (culture in 95% nitrogen, 5% CO_2_), hypoxia-mimetics such as dimethyl-oxaloylglycine (DMOG) or cobalt chloride (CoCl_2_), or by nutrient deprivation [65]. Further in vitro experiments showed that ERRα regulates angiogenic gene expression through promoter recognition in C2C12 myotubes, and pointed out that ERRα’s expression was HIF-1-dependent [65], for which the authors predicted 12 putative *HIF1A::ARNT* response elements in the *ESRRA* gene promoter [65]. Altogether, these authors suggest that HIF is involved in the hypoxic induction of ERRα in the skeletal muscle through the transcriptional regulation of ERRα expression. However, ERRα’s activity was not explored under HIF-1 depleting or activating conditions.

### 4.2. ERRα in Brain and Spinal Cord Hypoxia/Ischemia

Studies with astrocytes treated with CORM2, a CO-releasing compound that imitates ischemic brain injury, showed that ERRα/PGG1α can increase VEGF expression independent of HIF-1 (that is, even in HIF-1α-deficient cells) [67]. The treatment induces Heme Oxygenase-1 (HO-1) expression and its metabolites (CO and bilirubin) and promotes Ca^2+^ influx through L-type Ca^2+^ channels producing CaMKKβ-mediated AMPKα activation [67]. AMPKα increases NAMPT expression and NAD+ synthesis, which in turn increases SIRT activity. PGC-1α can be deacetylated by SIRT1 [67], and once deacetylated it interacts with ERRα to increase mitochondrial biogenesis and oxygen consumption [67] (Figure 3). With this model, the authors previously suggested that oxygen consumption aggravates intracellular hypoxia, allowing HIF-1α stabilization that further increases ERRα/PGG1α expression [60]. Using ChIP assays, the authors proposed that HIF-1 can stimulate ERRα’s transcription by binding to putative HIF-1 response elements (+539 to +542, 5′-CGTG-3′) within the promoter region of the ERRα gene [60]. HIF-1α knockdown blocked ERRα’s expression but not PGG1α’s in that HO-1 inducing model. Therefore, it is likely that HO-1 can stimulate VEGF both via HIF1-α dependent and independent mechanisms, the latter involving PGC-1α/ERRα and calcium regulation through the Ca^2+^/CaMKK/AMPK pathway [67] (Figure 3). These authors also propose some reciprocal and dynamic coordination between HIF-1α, and PGC-1α/ERRα for VEGF expression in astrocytic ischemia, involving mitochondrial biogenesis [60].

Other processes such as spinal cord injury (SCI) can manifest with ischemia, which aggravates secondary injury and neurological dysfunction [68,69,70]. Therefore, the vascular response is critical for SCI repair and includes HIF-1α and VEGF expression. In an SCI rat model, ERRα inhibition with XCT790 decreased VEGF and angiopoietin-2 expression [71], which in turn decreased endothelial cell proliferation, vascular density and produced histopathological changes to the spinal cord, such as inflammatory cell infiltration, hemorrhage and vacuolation, and fewer normal neurons, suggesting that ERRα activity is essential for SCI repair, in part by favoring adequate re-vascularization via VEGF [71]. In this model, it has not been explored whether ERRα‘s effects require HIF-1.

In the microglial cell line, BV2, pharmacological ERRα inhibition (with XCT790) or activation (with pyrido [1,2-α]-pyrimidin-4-one) were explored in combination with CoCl_2_ to mimic the hypoxia that accompanies SCI. Hypoxia induced HIF-1α and autophagy. ERRα’s effects were similar with/without hypoxia although more pronounced in hypoxia. During hypoxia, ERRα inhibition increased autophagy markers and increased IL-6, TNF-α and IL-10 mRNAs, but decreased FNDC5 (fibronectin type III domain containing protein 5). In turn, ERRα’s activation decreased p38 MAPK phosphorylation. The authors suggest that ERRα helps maintain homeostasis in microglia during hypoxia by down-modulating autophagy and inflammation [72].

### 4.3. ERRα in Hypobaric Hypoxia

On the other hand, in a non-pathological process such as exposure to high altitude, the expression of ERRα and PGC-1α are downregulated and the cell suffers mitochondrial dysfunction [73]. Treatment with dexamethasone maintains ERRα and PGC-1α levels similar to normoxia. This effect allows adaptability to hypobaric hypoxia in part through the expression of ERRα transcripts of mitochondrial dynamics proteins Fis1, Drp1 and Mfn2, that in turn increase OXPHOS [73]. This suggests that ERRα-mediated protection of mitochondrial bioenergetics is required for adaptation to hypobaric hypoxia.

### 4.4. ERRα’s Role in Cancer-Related Hypoxia

In parallel, ERRα has been extensively studied in solid tumors where blood vessels frequently become limiting to irrigate the tumor mass, leading to hypoxia. Thus, cancer represents another model where extensive evidence points to ERRα’s contribution to the hypoxia response. Cancer cells in solid tumors use typical HIF-1 orchestrated mechanisms to survive hypoxia [74], impacting angiogenesis, cancer stem cell maintenance, metabolic reprogramming, epithelial–mesenchymal transition (EMT), invasion, metastasis and resistance to therapy (radiation and chemotherapy) [74,75,76].

Simultaneously, the overexpression of ERRα has been associated with tumor aggressiveness and poor prognosis [77,78,79]. In 2002, Ariazi et al. suggested ERRα as a biomarker of unfavorable clinical prognosis in breast cancer, due to increased ERRα mRNA levels in primary tumor cells compared to normal mammary epithelial cells. ERRα’s expression correlated with high Her2/ErbB2, a tyrosine kinase receptor amplified in 15% to 25% of breast cancers that also confers aggressiveness [77] and that increases ERRα’s transcriptional activity via phosphorylation through MEK/MAPK and PI3K/Akt [79,80] (Figure 1).

Subsequent immunohistochemical analyses, mRNA quantification and gene expression profiles in several solid tumors (breast, cervix, colon, endometrium, ovary and prostate) are in agreement with Ariazi et al. and relate ERRα overexpression to cancer aggressiveness, increased risk of recurrence and lower survival [9,76,81,82,83,84,85].

In breast and prostate cancer, ERRα has been found to interact directly with HIF-1 with two main effects: (1) HIF-1 stabilization; and (2) an increase in the HIF-dependent expression of hypoxic genes [85,86]. This evidence has led to the suggestion that the direct ERRα-HIF interaction is another important mechanism by which ERRα contributes to the hypoxic response. The physical interaction between HIF and ERRα has been explored using a series of ERRα truncation mutants covering the N terminus, DBD, and LBD in GST pull down assays. Ao et al. suggested that the ERRα’s DBD is involved in HIF binding [86]. Anti-ERRα immunoprecipitation of MDA-MB-435 breast cancer cellular lysates treated with the iron chelator dipyridyl (DP) to stabilize endogenous HIF-1α, showed that all three ERRs associate to HIFα/β heterodimers both in vitro and in vivo, and this was abolished in ERRα mutants or with ERRα inhibitors [86]. Subsequent studies using co-immunoprecipitation and FRET in prostate cancer cells confirmed that the interaction happens and that it increases HIF-1′s transcription, but disagree on the ERRα domains involved [85]. Zou et al. suggest that the domain required for interaction with HIF-1 is the AF-2 region in ERRα’s LBD [85]. These authors further suggest that ERRα’s co-activator PGC1α may be necessary for its interaction with HIF-1. These effects are prevented in ERRα knockdowns or with ERRα’s inverse agonist XCT790 [85,86]. Evidence in prostate cancer cells suggests that the ERRα/HIF-1α interaction reduces the proteosomal degradation of HIF-1α [85]. These authors suggest that ERRα overexpression stabilizes HIF-1α and enhances HIF-1 transcriptional activity even under normoxia, with these effects amplified in hypoxia, resulting in a mechanism for the pre-adaptation to hypoxia [85].

In parallel, Stein et al. reported that ERRα regulates VEGF expression in breast cancer cell lines [61], similar to what was described in the previous section in angiogenesis models. A modified PGC1α that only binds to ERRα was used to induce VEGF expression in MDA-MB231 and MCF7 breast cancer cells, and this effect was abolished with ERRα knockdowns. A main ERRE was located within the transcribed region of the *VEGF* gene [61]. The positive regulation of VEGF by ERRα has also been observed in human breast tumors and in murine models [62], supporting that VEGF is a direct transcriptional target of ERRα in cancer, as in other cell types.

In summary, ERRα overexpression enhances the hypoxia response in solid tumors. It is likely that ERRα functions as an aggressiveness factor in cancer because it prepares cancer cells to resist metabolic stress and hypoxia. ERRα has been observed as active in immunosuppressive and immunoresistant tumors [87]. Cancer models have pointed to HIF-dependent mechanisms such as the physical interaction between ERRα/HIF-1α, as well as HIF-independent mechanisms such as direct VEGF modulation by ERRα.

### 4.5. ERRα and Kidney Hypoxia

Organs with high energy demand such as the brain, heart and kidney have low tolerance to hypoxia and are good models to evaluate ERRα’s effects. Physiological oxygen gradients across the renal cortex and medulla participate in the mechanisms to concentrate urine [88,89]. The healthy human kidney cortex presents around 50 mmHg of oxygen pressure, while the medulla has much lower oxygen pressures between 10 and 20 mmHg [88]. Keppner et al. recently evaluated the transcriptome during hypoxia (24 h at 0.2% O_2_) of the cortical kidney murine cell line mCCD(cl1) [89]. They found over 3000 differentially expressed genes, many related to aerobic metabolism and ATP production through mitochondria, and the hypoxia response was mainly driven by HIF-1 and not HIF-2. Interestingly, they knocked down ERRα and identified a reduced expression of some genes that typically function in hypoxia, such as *EGLN3* (an alpha-ketoglutarate dependent hydroxylase that controls cell proliferation and transcription upon hypoxia) and *SERPIN1* (plasminogen activator inhibitor-1, involved in the control of blood clotting) [89]. Since this regulation happened without a change in HIF-1α, the model suggests that ERRα controls the expression of specific genes important for the hypoxia response.

### 4.6. Hypoxia in the Invertebrate Fly Model

*D. melanogaster* is tolerant to oxygen starvation and can survive hypoxia for long periods of time. As in humans, the hypoxia response is importantly mediated by HIF (called sima in *D. melanogaster*); thus, *D. melanogaster* has been a study model for hypoxia [90] and represents an invertebrate model with a recognizable ERR. Li et al. showed that, in addition to HIF, the single ERR present in flies (called dERR) is necessary for the hypoxic response in *D. melanogaster*, since less than 25% of dERR mutant flies survived hypoxia [91]. Using single and double dERR and dHIF-1α mutants, they described genes sets that are important for hypoxia response and detected a subset of 282 dERR-dependent transcripts that are HIF-independent and whose expression changed in hypoxia, such as Pgi, Pfk, GAPDH2 and LDH [90]. This work suggests that the dERR has a prominent HIF-independent role in hypoxia adaptation, particularly via the upregulation of glycolytic enzymes.

Additionally, dERR was found to bind dHIF and participate in the HIF-mediated expression of its subset of genes [91]. The binding was shown by two hybrid screen and GST pull-downs and required dHIF’s residues 1289–1293 (LKNLL) and dERR’s LBD [91], in accordance to what Zou et al. described in cancer cells [85].

## 5. Conclusions

HIF-1 is the transcription factor usually considered the main regulator of the hypoxia response. However, ERRα, a cellular metabolism regulator, also plays a key role in hypoxia survival, in models ranging from invertebrates to vertebrates and in physiological and pathological scenarios. ERRα’s functions in hypoxia in most models include two mechanisms: (1) direct ERRα/HIF-1 interaction, which enhances HIF-1′s transcriptional activity at HREs (possibly without ERRα’s direct interaction with DNA); and (2) transcriptional activation by ERRα of genes that are classical HIF-1 targets, such as VEGF or glycolytic enzymes. The second mechanism can even happen in a HIF-1 independent manner that depends on ERREs coexisting with HREs.

ERRα is thus gaining recognition for its prominent role in the hypoxia response, both in the presence and absence of HIF-1. In many models, ERRα prepares cells for hypoxia, with important clinical/therapeutical implications and perspectives that could allow for the manipulation of tissues so they are pre-adapted to resist hypoxia or where ERRα is inhibited to hinder this adaptation. This is important, as hypoxia is central to numerous diseases with significant human mortality and high costs, such as cancer, cardiovascular and pulmonary disease, stroke, bacterial infections, inflammation, disorders related to prematurity and wound healing.

ERR’s expression and activity are conserved from Urochordates to mammals, suggesting that the ERR-mediated response to hypoxia appeared early in evolution. Phylogenetic exploration of the ERR-HIF interaction warrants more interrogation, with the potential to yield insights into its mechanisms and how they evolved.

Despite the many models that have described ERRα as responding to ischemia, it is unknown if ERRα may directly sense oxygen. No mechanism for direct oxygen sensing by ERRα has been described. Alternatively, ERRα’s activation upon ischemia/hypoxia may arise from the metabolic signals derived from ischemia or through HIF-1 stimulation. Protein tyrosine phosphatase 1B (*PTP1B*) and Parkin have also been shown to, respectively, decrease and increase the transcriptional activity of ERRα in hypoxia models (pancreatic islets [92] and HeLa cells [93]); thus, other pathways and layers for ERRα modulation likely exist.

## Figures and Tables

**Figure 1 ijms-24-07983-f001:**
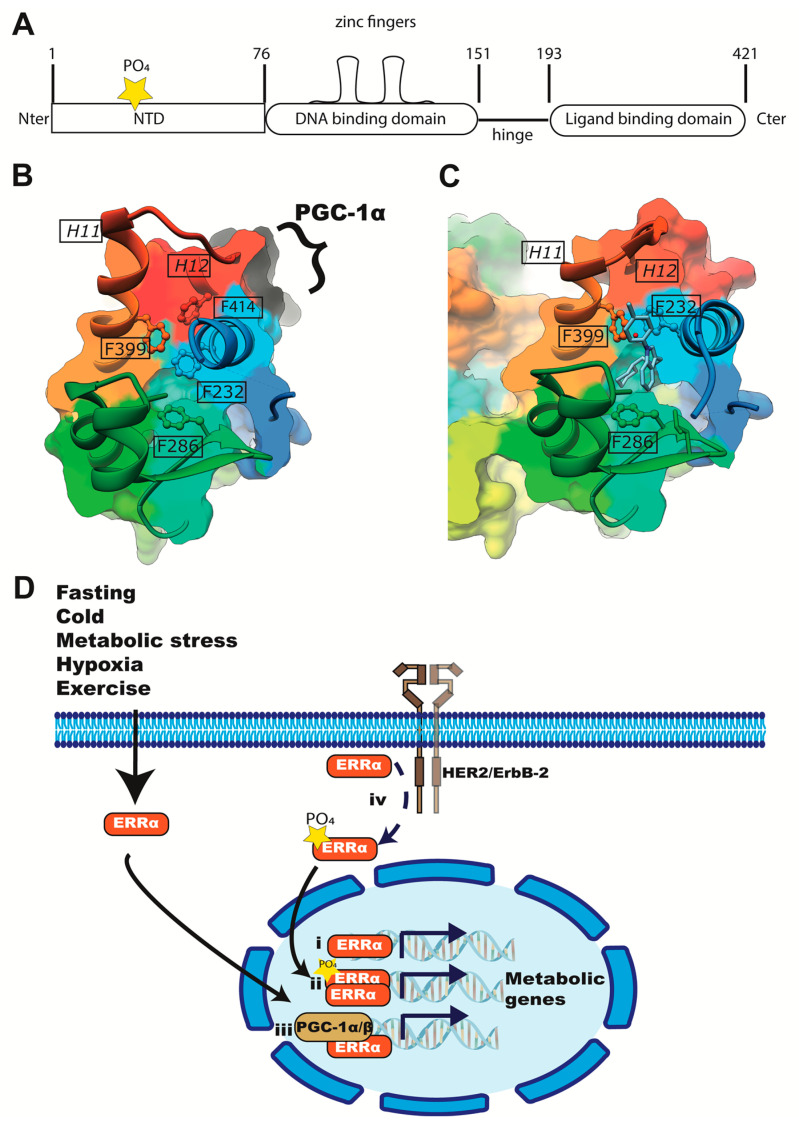
ERRα’s structure and function. (**A**) The general domain topology of ERRα and other estrogen-related receptors (NR3) includes an amino-terminal domain (NTD), a DNA binding domain, a hinge, and a putative ligand-binding domain (LBD). (**B**,**C**) ERRα’s LBD has been crystalized in the absence (**B**) and presence of ligand (**C**). The binding site is lined by residues F232, F286, F399 and F414 (PDBIDs 1XB7 and 2PJL) providing bulky side chains that fill the ligand-binding pocket in the absence of ligand (**B**). Moreover, without ligand, helix 11 (H11) lifts away from the binding site; helix 12 (H12) is perpendicular to H11 and residue F414 occludes the binding site (**B**). (**D**) ERRα can bind DNA as a (i) monomer or (ii) dimer. ERRα’s transcriptional activity increases in complex with co-activators, for example PGC-1 α or β (iii), and through posttranslational modifications, such as phosphorylation (iv), mediated by HER2-EGF. ERRα activity increases in metabolic stress, cold, fasting/nutrient deprivation, exercise and hypoxia.

**Figure 2 ijms-24-07983-f002:**
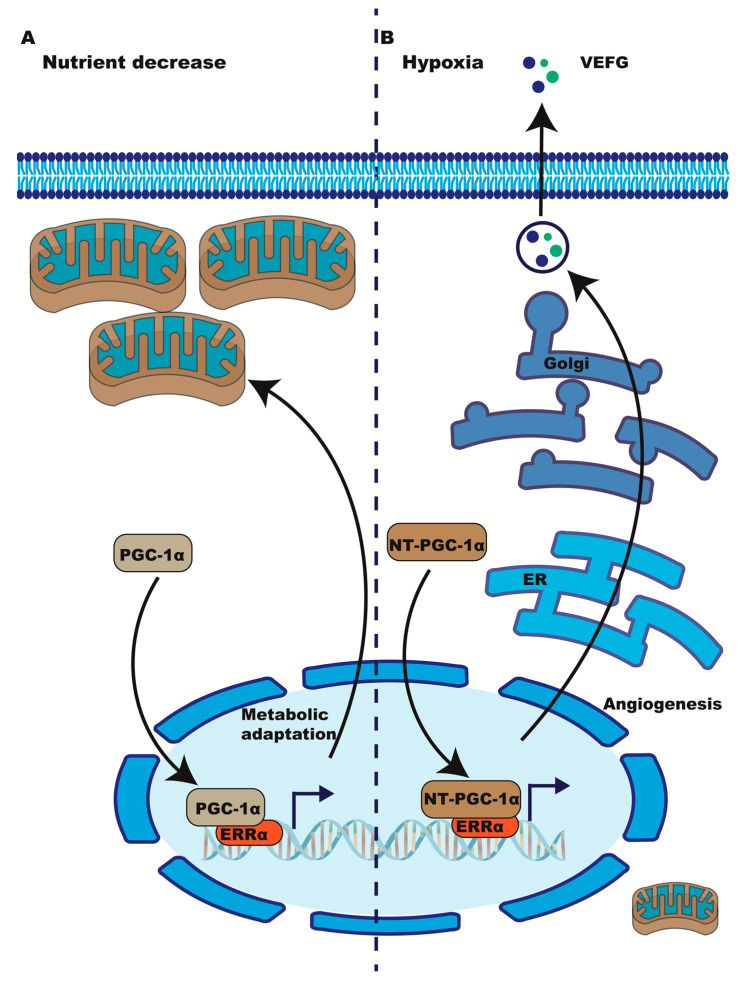
Metabolic adaptation via ERRα/PGC1α in normoxia (**A**) vs. hypoxia (**B**). (**A**) In normoxic nutrient deprivation, ERRα/PGC1α activate mitochondrial biogenesis. (**B**) In hypoxia, PGC-1α’s truncated isoform NT-PGC-1α binds ERRα but prevents the engagement of other transcription factors, limiting mitochondrial biogenesis and favoring the expression of angiogenesis genes such as VEGF. Oxygen consumption by mitochondria favors local hypoxia with concomitant HIF-1α stabilization.

**Figure 3 ijms-24-07983-f003:**
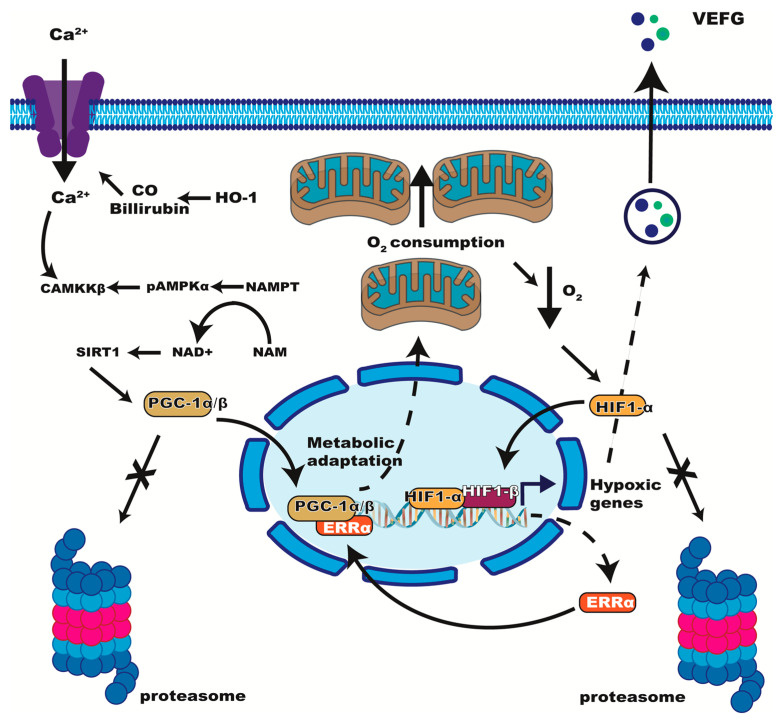
Proposed interactions between HIF-1 and ERRα pathways in hypoxia. ERRα/PGC-1α express hypoxia genes acting with HIF-1 or independent from HIF-1. Ca^2+^ activates CaMKKβ and AMPKα, increasing SIRT. PGC-1α is activated by SIRT1 deacetylation, decreasing its destruction through the proteasome.

## Data Availability

All the material pertinent to this review is cited in the references. No datasets or other forms of data were generated.

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
