# Peer review of "The Role of the Estrogen-Related Receptor Alpha (ERRa) in Hypoxia and Its Implications for Cancer Metabolism"

_ijms, 2023, doi:10.3390/ijms24097983_

Round 1
Reviewer 1 Report
Hypoxia is a fascinating and important topic because it is fundamental for understanding the tumor microenvironment and a crucial factor for tumor development. Hypoxia is the niche in which Cancer Stem Cells reside, and it is the stressor responsible for inducing the production of Circulating Tumor Cells. Hypoxia, moreover, is responsible for the increased tumor's radioresistance and chemoresistance.
The authors in this review make an exciting overview of hypoxia and its implication in tumor metabolism and explore in detail the function of ERRalpha in combination with HIF1.
The review is well-written and structured. It reads easily and pleasantly. The information provided is a lot and be connected.
However, there are some changes that I would suggest:
1) It is crucial in a review to make the understanding of the figures straightforward, and the figure itself must give all the information the reader needs. For example, in figure 3, it is possible to see colored circles and triangles, but their representation is unclear.
I would therefore modify all the figures by adding a legend.
2) In a review of hypoxia that can be considered comprehensive, the role of HIF2 cannot be ignored. HIF1 and HIF2 are essential to mark the boundary between acute and chronic hypoxia.
HIF1 and HIF2 activate different genes in cells at different times. Although in the early stages of hypoxia, there is an increase in HIF1 and HIF2 (acute hypoxia), after hours or days in some cells, the amount of HIF1 decreases (is degraded), and only HIF2 remains active (chronic hypoxia).
In the context of this review then, it would be good to mention the function of HIF2 in combination with ERRalpha, or at least devote a paragraph to HIF2 underling the difference with the HIF1 and clarifying the differences between acute and chronic hypoxia.
In conclusion, I would endorse the review after minor revisions.
Author Response
First, we apologize for the delay in responding the comments. We put a lot of effort into improving the manuscript.
We thank the reviewers for all their comments as we have found them helpful to improve the manuscript. We believe the current version has been improved it significantly. In the following lines, we will address their comments directly.
Reviewer 1:
1) It is crucial in a review to make the understanding of the figures straightforward, and the figure itself must give all the information the reader needs. For example, in figure 3, it is possible to see colored circles and triangles, but their representation is unclear.
I would therefore modify all the figures by adding a legend.
Answer: we have redone figure 1 through 3 emphasizing clarity, making identifying labels available in figure and rewriting the legends for each figure. We also replaced the original figure 1 with a new figure detailed ERRα domain structure as well as the binding site cavity. We believe the figures are significantly improved.
2) In a review of hypoxia that can be considered comprehensive, the role of HIF2 cannot be ignored. HIF1 and HIF2 are essential to mark the boundary between acute and chronic hypoxia.
HIF1 and HIF2 activate different genes in cells at different times. Although in the early stages of hypoxia, there is an increase in HIF1 and HIF2 (acute hypoxia), after hours or days in some cells, the amount of HIF1 decreases (is degraded), and only HIF2 remains active (chronic hypoxia). 

In the context of this review then, it would be good to mention the function of HIF2 in combination with ERRalpha, or at least devote a paragraph to HIF2 underling the difference with the HIF1 and clarifying the differences between acute and chronic hypoxia.
Answer: we have added two new paragraphs one of them describes the characteristics of HIF-2a and the other those of HIF-3a. As suggested, we emphasized the contrast in synthesis, degradation, and organ distribution of these other members of the HIF family.

Reviewer 2 Report
In this review article, the authors summarized the role of estrogen related receptor alpha (ERRa) in hypoxia. They first reviewed HIF1 is induced by hypoxia to promote oxygen delivery by inducing angiogenesis and erythropoiesis. Next, they explained the general characteristics of ERRa as a transcription factor, orphan nuclear receptor. They further summarized the interaction between HIF1 and ERRa pathways in the hypoxia response. Importantly, ERRa is induced by HIF1 at transcription level. Both ERRa and HIF1 functions cooperatively to activate angiogenesis or OXPHOS. They also reviewed the role of ERRa in cancer as a biomarker of cancer progression.
As the role of ERRa is important in several pathological conditions, especially cancer, this review will be interesting for many researchers. However, I think several points should be added or shown more clearly.
1) Genome wide analysis of transcriptome is important for analyzing the function of transcription factors. I wonder whether genome wide experiments such as ChIP-seq or RNA-seq have been conducted to find target genes of ERRa, although the authors showed several target genes of HIF1 and ERRa (Ln 71-85, 133-140, 207-209.).
2) The regulation of ERRa by hypoxia is not clearly described. The authors showed HIF1 binding sequence in ERRa promoter (Ln. 238), suggesting the indirect activation of ERRa by hypoxia. I wonder whether direct activation of ERRa by hypoxia has been reported.
3) Although the authors showed the involvement of ERRa in cancer progression, activation of mitochondria by ERRa might be important in the development of other diseases such as neurodegenerative diseases or metabolic diseases. Please summarize the association between ERRa depletion and these pathological conditions.
4) Several sentences should be rephrased because it is difficult to understand.
Ln. 215 In vitro experiments pointed out that ERRa’s expression is HIF-1 is activity dependent.
5) Ln. 253 ERRa inhibition with XCT790, in a SCI murine model, decreased the expression of these proteins.
It is unclear what proteins “these proteins” indicates. Please rephrase.
Author Response
First, we apologize for the delay in responding the comments. We put a lot of effort into improving the manuscript.
We thank the reviewers for all their comments as we have found them helpful to improve the manuscript. We believe the current version has been improved it significantly. In the following lines, we will address their comments directly.
Reviewer 2:
- Genome wide analysis of transcriptome is important for analyzing the function of transcription factors. I wonder whether genome wide experiments such as ChIP-seq or RNA-seq have been conducted to find target genes of ERRa, although the authors showed several target genes of HIF1 and ERRa (Ln 71-85, 133-140, 207-209).
Answer: We included in this review several ChIP-seq studies published recently and summarized them in paragraph XXX.
2) The regulation of ERRa by hypoxia is not clearly described. The authors showed HIF1 binding sequence in ERRa promoter (Ln. 238), suggesting the indirect activation of ERRa by hypoxia. I wonder whether direct activation of ERRa by hypoxia has been reported.
No direct activation has been reported to date, only that ERRa can be activated independently of HIF-1a.
3) Although the authors showed the involvement of ERRa in cancer progression, activation of mitochondria by ERRa might be important in the development of other diseases such as neurodegenerative diseases or metabolic diseases. Please summarize the association between ERRa depletion and these pathological conditions.
To the best of our knowledge, no correlation/association has been put forward relating ERRa and neurodegenerative diseases. However, a great review of the regulation of metabolisms by ERRs on a murine model is cited in the text.
4) Several sentences should be rephrased because it is difficult to understand. Ln. 215 In vitro experiments pointed out that ERRa’s expression is HIF-1 is activity dependent.
5) Ln. 253 ERRa inhibition with XCT790, in a SCI murine model, decreased the expression of these proteins. It is unclear what proteins “these proteins” indicates. Please rephrase.
We have improved the writing across the manuscript to improve clarity.
We hope these changes are enough to give our manuscript the quality required to publish it.

Round 2
Reviewer 2 Report
The authors responded satisfactorily to my concerns.